# Determination, Separation and Application of ^137^Cs: A Review

**DOI:** 10.3390/ijerph191610183

**Published:** 2022-08-17

**Authors:** Yiyao Cao, Lei Zhou, Hong Ren, Hua Zou

**Affiliations:** Department of Occupational Health and Radiation Protection, Zhejiang Provincial Center for Disease Control and Prevention, Hangzhou 310051, China

**Keywords:** ^137^Cs, environment, radioactivity, determination, application

## Abstract

In the context of the rapid development of the world’s nuclear power industry, it is necessary to establish background data on radionuclides of different samples from different regions, and the premise of obtaining such basic data is to have a series of good sample processing and detection methods. The radiochemical analysis methods of low-level radionuclides ^137^Cs (Cesium) in environmental and biological samples are introduced and reviewed in detail. The latest research progress is reviewed from the five aspects of sample pretreatment, determination, separation, calculation, application of radioactive cesium and the future is proposed.

## 1. Introduction

^137^Cs (T½= 30.17 y) is among the most important hazardous radionuclides involved in radiological pollution to humans and the environment due to its long half-life and high fission yields [1,2]. ^137^Cs, as one of more than 30 isotopes of cesium, is fission a product of ^235^U, whose decay chain is shown in Figure 1 [3,4]. ^137^Cs decays to short-lived ^137m^Ba (T½ = 2.61 m) via emitting beta rays at maximum energies of 512 (94.0%) and 1176 keV (6.0%), respectively. ^137m^Ba emits gamma rays with an energy of 661 keV and becomes stable ^137^Ba [5]. There are three main sources of ^137^Cs in the environment: (1) Global fallout from the atmospheric nuclear weapon tests during 1950s–1980s [6,7,8,9], with total inventory of 545–765 PBq ^137^Cs [10,11,12]. (2) Nuclear accidents, including the former Soviet Union Chernobyl nuclear power plant accident in 1986 and the Fukushima Daiichi nuclear power plant accident in 2011 in Japan. The Chernobyl accident released 85 PBq ^137^Cs into the surrounding environment, especially in the Baltic Sea, while the Fukushima accident was reported to release 15–20 PBq ^137^Cs mostly into the Pacific Ocean [13,14]. (3) Regulated releases from the operation of nuclear facilities, such as nuclear power plants and nuclear reprocessing plants [15]. 

With a chemical similarity to K^+^, extensive investigations revealed that cesium is prone to be transferred into the human body via the food chain and substitute for potassium during transport in the cell membrane [16]. ^137^Cs has serious damaging effects on the human body due to the gamma radiation from its daughter ^137m^Ba. ^137^Cs is mainly accumulated in bone and muscle tissue, thereby can induce soft tissue tumors to cause cancer, such as thyroid cancer, ovarian cancer, breast cancer, bladder cancer, and bile duct cancer [17,18]. The chronic damage of ^137^Cs to the human body also manifests as inflammatory lesions of various tissues and organs, the most obvious of which is inflammation of the lungs, gastrointestinal tract, urinary tract, and reproductive system. Therefore, it is vital to remove cesium, especially after the Chernobyl disaster and the Fukushima Daiichi nuclear plant accident, where large amounts of ^137^Cs were released into the environment. On the other hand, the separation of long-life radionuclides from aqueous solutions can greatly reduce the time required for safe storage for natural decay [19], and various functional adsorbents have made great contributions to the efficient removal of ^137^Cs in the environment.

As fission products with well-documented source terms, ^137^Cs have been applied as environmental tracers for studying ocean circulation and sedimentation processes, etc. [9,10,11]. With the rapid development of the global nuclear power industry and the widespread application of nuclear technology, it sets high demands for environmental radiation safety and radiological risk assessment during routine operation and nuclear emergency situations. Therefore, it is of great significance to establish efficient methods to be applied for the determination and enrichment of ^137^Cs. This article summarizes the recent progress of ^137^Cs sample pretreatment, measurement, separation, evaluation of measurement data, and application.

## 2. Sample Pretreatment

Pretreatment is required before the sample can be analyzed. The purpose of sample pretreatment is to obtain a homogeneous sample solution and preliminarily remove sample matrix and interferences, such as organic matter and stable elements. Obviously, sample pretreatment is a crucial stage before analysis. The selection of sample pretreatment methods depends on the sample type and the analyte to be analyzed. Herein, we divide the sample type into three categories for discussion, including environmental solid samples, environmental water samples, and biological samples.

### 2.1. Pretreatment of Environmental Solid Samples

Environmental solid samples include soil, sediment, sludge, etc. The commonly used methods for soil samples are acid leaching with microwave digestion or not [7,8,20,21,22,23]. Generally, the sample needs to be dried and combusted at 450–600 °C to decompose organic matter. Choi et al. [24] found that the volatilization of ^137^Cs did not occur until 500 °C, and the volatilization of cesium in milk samples during ashing did not occur until 450 °C. In order to avoid the loss of Cs during the combustion process, staged ashing can be used [25], i.e., ashing the samples at about 400 ℃, for Cs determination. Sample pretreatment for ^137^Cs determination by gamma spectrometry do not need acid digestion.

Another method is alkaline fusion [26,27,28]. Jurecic et al. [29] compared conventional acid digestion using a mixture of HNO_3_, HClO_4_, and HF acid, microwave dissolution using HNO_3_ and HF, and alkali fusion with Na_2_CO_3_ and Na_2_O_2_. Complete decomposition of soil samples can only be achieved by alkaline fusion. The boric acid melting method can be used to dissolve Cs samples. Taylor et al. [28] mixed soil sample with LiBO_2_ and LiI in a platinum crucible and melted it at 900 ℃, then used nitric acid to dissolve. The borate melting method can maintain a relatively high recovery rate of Cs.

### 2.2. Pretreatment of Environmental Water Samples

For environmental water samples including seawater, groundwater, lake water, and river water, after acidification and concentration treatment, ^137^Cs can be enriched by cation exchange resin [20,30,31]. The sediment in water needs to be decomposed by ashing. Firstly, the main purpose of acidification is to eliminate interference [32]. Secondly, concentration treatment can use precipitation or evaporation. For ^137^Cs, the seawater sample was acidified to pH 1.6 with HNO_3_, CsCl was added and then adsorbed with ammonium phosphomolybdate (AMP).

### 2.3. Pretreatment of Biological Samples

For the pre-treatment of biological samples, including bones, teeth, blood, urine, as well as plants, milk, etc. The solid biological samples use similar acid digestion methods as applied for environmental solids can be adopted [7,21,22,33,34]. For example, Gasa et al. [33] used thermal HF, HNO_3_, HCl, and H_3_BO_3_ to digest mammalian skulls. Because the teeth and bones are rich in Ca, the fuming nitric acid method is also commonly used for pretreatment [35]. In the report of Altzitzoglou et al. [36], the bone ash is almost completely dissolved by boiling HNO_3_ and H_2_O_2_ after repeated digestion and evaporation. After the last digestion, the evaporated residue was dissolved in boiling HNO_3_-Al (NO_3_)_3_.

The traditional method of processing milk samples is to store them in formaldehyde or sodium azide first [37,38] to maintain them in an antiseptic environment. One method is ashing at high temperatures to remove a large amount of fat and protein, then the ash is dissolved in concentrated nitric acid, but this method takes a long time. Thus, generally, other methods are used, such as ion exchange resin for the enrichment of radioactive elements, cation exchange resin, such as DOWEX 50W-X8 [39,40], anion exchange resin, such as DOWEX 1-X8 [39,41]. In addition to the above two methods, microwave digestion can also be used. Branislava et al. [42] placed aliquots of milk samples into digestion vessels, added concentrated nitric acid and 30% hydrogen peroxide and heated them at 900 W for 25 min. Urine samples are usually digested by heating with concentrated acid. In recent years, cation exchange resins have been developed to remove the matrix. For example, pigments in urine can be removed by the combination of activated carbon and ion exchange resin. 

## 3. Measurement

As mentioned above, ^137^Cs decay into ^137m^Ba through β emission, and immediately accompanied by 662 keV gamma-ray emission, ^137^Ba was formed. In addition, due to the low concentrations in environmental samples of most of the long-lived radioisotopes of interest in environmental monitoring, such as the isotopes of cesium, measurement approaches require extremely high sensitivity, and various detection methods have been developed [43]. Beta counting or gamma spectrometry can be used to detect ^137^Cs, but gamma spectrometry is more commonly used because most samples can be directly counted without any chemical separation. The development of mass spectrometry technology has led to its application in more and more radionuclide measurements, and mass spectrometry can measure multiple nuclides simultaneously.

### 3.1. γ-Ray Spectrometry of ^137^Cs

Gamma-ray spectrometry is an analytical method that allows the identification and quantification of gamma emitting isotopes in a variety of matrices. The sample with a certain geometric shape is placed in the appropriate position of the germanium or sodium iodide detector of the spectrometer system, the sample γ-ray spectrum is obtained to determine the position of the full-energy peak and the net peak area, γ-ray emission probability, sample mass (or volume), and related parameters or correction coefficients, etc. to determine the type of radionuclide contained in the sample and its specific activity. **γ** spectrometer consists of preamplifier, amplifier, pulse amplitude analyzer, high voltage power supply, spectrum data analysis, and processing system, etc. Gamma spectroscopy is convenient and fast, and does not require complex sample preparation. It is sufficient for the ashing process of most solid samples [7,33,44], and the detection limit of high purity germanium (HPGe) can be as low as ~1 mBq/g [45]. Indeed, gamma rays are slightly attenuated by the sample matrix, meaning that direct measurement can be carried out [46]. Therefore, it has become the most widely used ^137^Cs measurement method [47,48].

### 3.2. Mass Spectrometric Techniques for the Determination of ^135^Cs/^137^Cs

Highly sensitive mass spectrometric techniques are important methods for measurement of radionuclides, especially for long-lived radionuclides. Since the half-lives of ^137^Cs are relatively short, gamma spectrometry is suitable when only measuring the concentration of ^137^Cs separately is required. However, because ^135^Cs has a longer half-life of 2.3 × 10^6^ years and a low β energy of 76 keV, when measuring ^135^Cs / ^137^Cs, the analysis method based on mass spectrometry is the first choice [49]. After the chemical separation of the environmental sample, most of the interfering elements have been separated, but there will still be trace interfering elements in the final sample solution, and Table 1 shows the interferences affecting of ^137^Cs. Therefore, it is necessary to use the instrument to further suppress the signal of the interfering elements.

#### 3.2.1. Inductively Coupled Plasma Mass Spectrometry (ICP-MS)

For the determination of radionuclides, ICP-MS is the most widely used mass spectrometry technique due to its relatively low cost, easy operation, and high sensitivity. Compared with the long counting time of alpha spectrometry or beta counting, ICP-MS is more efficient and can be completed within a few minutes for multi-radionuclide measurements. Therefore, ICP-MS was used in the rapid separation of radionuclide traces in the environment and the separation of fission products and actinides in nuclear samples in the 1990s [50,51,52,53,54]. However, for ^137^Cs, the interference is mainly ^135^Ba and ^137^Ba. To solve this problem, in recent years, inductively coupled plasma-quadrupole mass spectrometry (ICP-QMS), sector-field inductively coupled plasma-mass spectrometry (SF-ICP-MS), dynamic reaction cell inductively coupled plasma mass spectrometer (DRC-ICP-MS), electrothermal vaporization inductively coupled plasma mass spectrometry (ETV-ICP-MS) and other methods have been developed [3,27,28,55,56]. 

In an ICP-QMS with a reaction cell, the reaction gas must react with the interfering substance, and it has little or no effect on the sensitivity of the element of interest. It can be selected based on thermodynamic and kinetic data [2,3,27,28,49,57]. For ^137^Cs, H_2_ and He can be introduced into the reaction cell to suppress the Ba isobaric line.

Inductively coupled plasma tandem mass spectrometer (ICP-MS/MS) has two quadrupole mass filters, equipped with a reaction cell in the middle of the two mass filters, which can effectively remove ^135^Ba and ^137^Ba in the reaction cell to test ^135^Cs/^137^Cs [26,50,55].

#### 3.2.2. Resonance Ionization Mass Spectrometry (RIMS)

The RIMS can avoid some isobaric interference and improve the selectivity of traditional mass spectrometry to isotopes [58]. The system is mainly composed of three parts: Ion source sample loading system, atomic ionization continuous wave laser system, and mass spectrometer [59]. In RIMS, a laser beam is used to selectively/resonantly excite and ionize analyte atoms, thereby eliminating isobaric interference caused by other elements. RIMS also has its unique advantages in the analysis of ^135^Cs and ^137^Cs. For example, adjusting the wavelength of Cs atomic radiation by a laser will result in higher sensitivity. The detection limit is about 10^6^ atoms [60]. The main limitation of RIMS is the high cost of complex instruments.

#### 3.2.3. Thermal Ionization Mass Spectrometry (TIMS)

The TIMS ion source is equipped with a sample changer, a sector magnetic field mass analyzer, and a Faraday cup detector [3,26,57]. By setting the magnetic field under the control of a dedicated personal computer, TIMS works in a peak-hopping mode, in which the specified ion beam is directed to the detector in sequence, the computer acquires voltage data in a digital format as a measure of peak intensity, and calculate the specified isotope ratio. The characteristic of TIMS is the abundance sensitivity of 10^−10^–10^−11^ at *m*/*z* 135, which is suitable for the analysis of ^135^Cs and ^137^Cs [61]. TIMS is widely used for nuclear safeguarding and forensic purposes due to providing high accuracy, precision, and sensitivity, additionally exhibiting the advantages of negligible spectral interference, matrix effects, and mass bias [62,63,64].

#### 3.2.4. Accelerator Mass Spectrometry (AMS)

AMS can also measure trace amounts of ^137^Cs, which has the characteristics of high sample throughput, high sensitivity, and fast turnaround time [49]. However, when measuring the atomic ratio of ^135^Cs/^137^Cs by AMS, the biggest challenge is to separate Ba and Cs under high energy conditions [65]. In addition, the test and analysis processes are complicated. AMS is similar to RIMS in that it is expensive, the installation process of the instrument is complicated, and the target preparation process of AMS is relatively time-consuming and laborious [66]. Consequently, by comprehensively comparing the characteristics of different mass spectrometry measurement analysis methods, ICP-MS is still the mainstream in the measurement of ^135^Cs and ^137^Cs. 

In the literature of the past 15 years, the pretreatment, separation methods, measurement techniques, chemical recovery, and detection limits for ^137^Cs are summarized in Table 2. 

## 4. Separation and Purification

The main purpose of chemical separation is to concentrate the target radionuclide and to remove interferes, thus obtaining a purified fraction. At present, a variety of separation methods have been applied, and many functional materials have been derived for the removal of radiocesium. In the following sections, we summarize the relevant separation methods and a series of novel environmental materials in detail.

### 4.1. Extraction Chromatography

Extraction chromatography combines the high selectivity of solvent extraction with the simple and multistage features of column chromatography, which requires simple operation, easier material handling and lower cost input. More importantly, extraction chromatography can significantly reduce the volume of highly radioactive solutions and the amount of solid waste, reducing the harmful effects of radioactive waste on human health and the ecological environment [78]. Russell et al. [27] tried a variety of extraction methods, AG 50WX8 cation exchange resin and Sr resin, organic extraction material (BOBCCalixC6) with linear alkyl ester as the skeleton, Amberchrom CG-71 resin, after adsorption can use HNO_3_ elutes Cs. This method can effectively reduce the operation steps and is suitable for the measurement of low-concentration environmental samples. The price of extraction resin is relatively high, but the experimental separation process is fast, the elution volume is small, and the amount of waste liquid generated is small. 

### 4.2. Ion Exchange

Ion exchange is a method that removes radioactive isotopes from the liquid phase, during which some toxic ions may be reduced while more harmless ions may be increased in the aqueous phase [79]. The ion exchange method can deliver high selectivity, minimal radioactive discharge, and solidified waste [80]. Many natural substances have an ion-exchange capacity for various radionuclides, such as clay minerals [81], zeolites [82], etc. Belviso et al. [83] utilized synthesized zeolites from volcanic ash as an alternative adsorbent material for cesium removal from aqueous solutions with encouraging results. However, the low ion exchange capacity of these natural minerals is limited in practical applications [84]. The ideal ion exchange material should have high adsorption or exchange capacity and be stable under a wide range of environmental conditions. Therefore, researchers carried out a series of modifications of natural substances to improve performance. Chen et al. [85] developed a novel montmorillonite-sulfur composite via a one-step solvent-free method and applied it to the removal of Cs^+^. Due to the stronger interaction between the soft Lewis base S^2−^ ligand and the soft Lewis acid Cs, compared with other cations, the capacity and selectivity towards Cs^+^ was significantly enhanced. The maximum adsorption capacity of the adsorbent reached 160.9 mg/g, and the distribution coefficient value (~4000 mL/g) was three times larger than that of the pristine montmorillonite (~1500 mL/g). 

In addition, other novel ion exchangers have also been developed. Metal hexacyanoferrate (MHCFs) is a typical coordination polymer consisting of transition metal ions (such as Cu^2+^, Co^2+^, Ni^2+^, and Fe^3+^) with coordinated CN bridges. Since their structural lattice is comparable to the size of hydrated cesium ions, MHCFs can selectively adsorb cesium ions through ion exchange [19]. At the same time, they should be easily regenerated and reused in an environmentally friendly way. Zhang et al. [86] prepared a clay-based composite hydrogel (DHG(Cu)) containing hexacyanoferrate (HCF) nanoparticles for selective removal of Cs^+^ from contaminated water. DHG(Cu) showed excellent recovery performance for Cs^+^ in a polluted environment, and the maximum adsorption capacity was 173 mg/g. In KCuHCF, the adsorption of Cs^+^ was mainly achieved through the ion exchange between Cs^+^ and K^+^. Unfortunately, due to the extremely small particle size (~10 nm) of this excellent material, its practical application is limited. 

Ammonium phosphomolybdate (AMP) is another novel ion exchanger and has a high selective adsorption capacity for Cs. The chemical formula of AMP is [(NH_4_) _3_PMo_12_O_40_]. AMP contains exchangeable NH_4_^+^, which can be replaced by Cs^+^. The adsorption coefficient (K_d_) of Cs is about 10^4^ mL/ g and is not significantly affected by the high ionic strength of the solution [2,27,87]. After concentrating cesium, AMP is usually dissolved with sodium hydroxide solution to produce a precipitate of cesium iodate bismuthate. At present, AMP has been widely used in the adsorption of Cs in samples such as seawater, inland freshwater, nuclear waste liquid, and digested soil and sediment. Hyoe et al. [70] compared the separation efficiency of 5 AMP manufacturers and confirmed that the AMP may contain Cs in the manufacturing process, and the AMP produced by Kishida Chemical Co., Ltd. (Osaka, Japan) had the least Cs content. On this basis, the adsorption efficiency of AMP and AMP-PAN, KNiFC-PAN to Cs, and interfering elements Ba, Mo, Sb, and Sn were further studied. By combining inorganic ion exchangers (AMP and KNiFC) with organic polymers (polyacrylonitrile, PAN), the granularity performance of powdered AMP and KNiFC can be improved.

### 4.3. Adsorption

Adsorption is a process of using a solid adsorbent to adsorb one or several components in a water sample on the surface and then desorb the predicted components by appropriate solvent or heating to finally achieve the purpose of separation and enrichment. The adsorption method has the advantages of low cost, excellent removal efficiency, high flexibility in operation, and no generation of secondary pollutants [88,89,90,91]. Thus, adsorption is considered to be the most effective method for removing environmental pollutants [92]. Various adsorbents, including metal sulfides, metal-organic frameworks (MOFs), graphene oxide (GO), etc., have been developed to remove cesium through different mechanisms. Because soft S^2−^ ligands in their frameworks have a strong innate affinity for soft metal ions (Cs^+^) rather than coexisting hard ions (H^+^, Na^+^, K^+^) [93], they can effectively remove radionuclides from wastewater. Xing et al. [94] utilized graphene oxide for the removal of cesium from aqueous solution, and the maximum adsorption capacity of GO could reach 95.46 mg/g due to the existence of a large amount of oxygen-containing functional groups.

However, the selectivity enhancement of adsorbents to target nuclides remains a great challenge. To address the challenges, the ion imprinting technique has been developed as a template strategy to effectively improve the selectivity of materials by which a functional monomer and a crosslinker were polymerized in the presence of a template ion [95,96]. It can create specific cavities containing a ligand for the template ion that possesses the right size and charge [97,98]. More and more ion-imprinted polymers are now available, and the selective adsorption of various nuclides can be achieved by introducing an imprinted cavity for the target nuclide on the surface of ion-imprinted polymers [99]. Zhou et al. [97] employed dual ion-imprinted mesoporous silica based on multiple interactions to separate cesium from a high-salt environment. The synergism of multiple interactions and imprinted cavity endows the sorbent with good selectivity for cesium over other cations and with excellent salt tolerance. 

## 5. Evaluation of Measurement Data

The complete measurement process should also include blank experiments, activity calculations, and precision calculations. Whenever reagents are changed and each batch of samples is analyzed, a blank experiment should be conducted. Under normal circumstances, the number of blank samples should not be less than 5% of the total number of samples analyzed. Finally, the repeatability and reproducibility of the method should be evaluated.

In order to evaluate the impact of various chemical operations and measurement procedures on data quality, relevant scholars conducted uncertainty analysis on different measurement methods in different environments. The measurement uncertainty of ^137^Cs in water with α and β measuring instruments mainly comes from the contribution of β radioactivity measurement with its inherently lower background, which is considerably more sensitive [100,101,102]. The main uncertainty of ^137^Cs measurement by the γ energy spectrum analysis method comes from the peak count of ^137^Cs in the sample source. In actual work, the sample volume can be selectively increased according to the specific experimental conditions, the detection efficiency can be improved, and the measurement time of the sample source can be extended to increase the peak zone count of ^137^Cs in the sample source [103].

In order to analyze the low-level ^137^Cs activity in complex samples, the instrument background level, counting efficiency, sample number, and element yield are critical. To reduce the errors introduced during the analysis. Cresswell et al. [104] analyzed the uncertainty and detection limit of ^137^Cs activity measured by the computer-based gamma-ray spectrometer (AGS) with the gross counts within the defined spectral windows method. The results show that the spectral interference stripping contributes the most to the uncertainty, so a narrow spectral window should be used. To use a narrow spectral window, it needs to control the spectral gain and resolution. Caciolli et al. [105] studied the full spectrum analysis (FSA) by using a sodium iodide scintillator. By introducing the non-negative least squares constraint into FSA algorithm, a more reliable basic spectrum was obtained, and the uncertainty in the fitting process was reduced. 

## 6. Tracer Application of ^137^Cs

Due to nuclear tests and nuclear accidents, radioactive materials are mainly present in sediments and oceans worldwide [11,56,75,106]. After the radioactive material is deposited on the ground, it is resuspended with the storm and the surface soil particles in the air, and deposited on the ground again. Under dry and wet climate conditions, the difference in the ^137^Cs/^90^Sr activity ratio in the topsoil provides clues for understanding the source of dust. Most of ^137^Cs exist in the seawater in dissolved form. Therefore, ^137^Cs nuclides can be used as a tracer for the study of surface dust suspension, transportation, and their deposition at a distance, as well as marine transportation and eddy current research for final assessment of nuclear tests and pollution status of nuclear accidents [13,107,108,109,110]. 

### 6.1. Application of ^137^Cs as Oceanographic Tracer

The Nielsen research group has studied the content of ^137^Cs and ^99^Tc in the Danish Strait seawater in the past 40 years [109], trying to reveal the water mixing dynamics of the Danish strait. ^99^Tc mainly came from the Chernobyl nuclear accident. After the accident, it spread evenly in the sea water of the Danish Strait. ^137^Cs were continuously discharged from the water treatment plant of the European nuclear power plant. Overall time-series records of ^137^Cs and ^99^Tc in the Danish Straits could serve well as oceanic tracers and ^99^Tc/^137^Cs activity ratio could be utilized to explore the flow mixing dynamics between the North Sea and the Baltic Sea.

The radioactive materials, ^134^Cs and ^137^Cs, produced by the FDNPP accident were mainly distributed in the North Pacific and its marginal areas in 2011. It is estimated that the total amount of these radionuclides released directly into the ocean is about 15–19 PBq. ^134^Cs has a short half-life of about 2.06 years, and the ratio of ^134^Cs / ^137^Cs is close to 1, so ^137^Cs has become the main detection target. Hirayama et al. [111] described a first attempt to estimate the water seepage rate of an active crater lake using radioactive cesium dispersed into the environment by the Fukushima Dai-ichi Nuclear Power Plant accident in March 2011 as a hydrological tracer and successfully estimated the water leakage rate of the Yugama crater lake in Kusatsu Bairen volcano. The findings suggest that radioactive cesium is one of the powerful tracers for addressing the geochemical processes of the Crater Lake System. Hirose [13,14] discussed the temporal changes of ^90^Sr and ^137^Cs in the surface waters of Japanese Haiti. In subsequent studies, the apparent vertical diffusion coefficients of ^90^Sr and ^137^Cs (AVDC) and apparent initial surface flux (AIF) show temporal and spatial variability. A mathematical model of its vertical distribution has been established. Zhou et al. [106] analyzed ^137^Cs, ^134^Cs, ^90^Sr, and total β in the northeastern South China Sea, Luzon Strait, and its vicinity. The peak of ^137^Cs appeared at a depth of 150 m, and the peak of ^90^Sr at 0.5 m in depth, the average value of total β activity gradually increased from the South China Sea to the Western Pacific. The highest average value of ^134^Cs and ^137^Cs was found at a depth of 500 m in the Northwest Pacific Public Area [75]. Liu et al. [56] analyzed the spatial and vertical distribution of Pu isotopes and ^137^Cs in the surface sediments and core sediment samples of the Yangtze Estuary. The activity of ^137^Cs in the surface sediment samples of the Yangtze Estuary ranged from below the detection limit (0.8 mBq/g) to 4.92 mBq/g, there is no obvious spatial change. The maximum peak of ^137^Cs activity of 19.55 ± 1.38 mBq/g in the depth range from 140 to 180 cm. 

### 6.2. Application of Global Fallout ^137^Cs as a Tracer for Soil Erosion Investigation

The classical approach of using ^137^Cs as a soil erosion tracer, which is based on the comparison between stable reference sites and sites affected by soil redistribution processes, has been applied to the research focused on the establishment of erosion rate calculation models, determination of regional background values, and spatial variability of soil erosion (erosion and subsidence rate, slope erosion process). The method is associated with potentially large sources of uncertainty, with major parts of this uncertainty being associated with the selection of the reference sites. Laura et al. [112] proposed an extension of this procedure, using a repeated sampling approach, in which the reference sites are resampled after a certain time period. Suitable reference sites are expected to present no significant temporal variation in their decay-corrected ^137^Cs depth profiles. Olson et al. [113] determine the soil erosion rates in cropland of west central Illinois using a magnetic tracer (fly ash) and radio-cesium (cesium-137). The fly ash and cesium-137 accumulation on a stable cropland/hayland summit was determined using a spiral transect. This reference site was used as a baseline and then compared with the fly ash and cesium-137 levels in adjacent cropland landscape positions to estimate loss from erosion. Li et al. [114] used the ^137^Cs tracer method to study the soil erosion of purple soil slope farmland in the Three Gorges Reservoir area of China, and calculated the soil erosion rate of slope farmland using an improved simplified mass balance model. The segmented analysis results of the soil erosion rate along the slope show that due to the effect of plowing, the soil erosion rate of the two slope sections of the slope farmland generally decreases with increasing slope length, and accumulation occurs below the slope section. Zhao et al. [115] Collected soil samples from sandy land, bare land, Gobi, cultivated land, and grassland in the Zhundong region of Xinjiang, measured the ^137^Cs content, and compared it with the soil erosion data of various types of land. In low-grain wind-eroded areas, ^137^Cs tracer technology is not reliable.

## 7. Conclusions

This article reviewed the measurement, separation, and applications of ^137^Cs in environmental samples. It also discussed in detail the pretreatment of different sample types, chemical purification methods, and analytical techniques, as well as their respective advantages and disadvantages. Pretreatment includes solids, water bodies, and biological samples. Chemical purification methods include precipitation, ion chromatography, and extraction chromatography. Analysis techniques include traditional beta counting and gamma-ray measurement, as well as the latest development of mass spectrometry. At present, the most widely used pretreatment method is acid hydrolysis. The chemical separation method is coprecipitation. Although its process is cumbersome, it is still the preferred method for the treatment of large numbers of samples. At the same time, crown ether extraction chromatography is gradually replacing the coprecipitation method because of its superior separation efficiency. 

The rapid analysis method of ^137^Cs still needs to be further explored. Traditional methods often take two weeks or more, and the standard processes established by most countries are still based on the results from the last century. To solve this problem, it is necessary to increase the selection of pretreatment, chemical separation, and test methods. In terms of pretreatment, the rapid processing of large volume samples still needs to be explored, especially the concentration of liquid samples. In terms of chemical separation, crown ether extraction chromatography has become a new trend. On the one hand, it is the synthesis of new crown ethers, which makes the decontamination factor and chemical recovery higher. Another aspect is how to achieve mass production. In terms of the selection of test methods, the advantages of mass spectrometry are obvious. The following work can be expanded from the following aspects. First, the combination of chemical separation and instrumental analysis to reduce contamination during operations, such as ion exchange, electrothermal evaporation, laser burning, or capillary electrophoresis for preliminary treatment before injection. Second, the common factor restricting RIMS, TIMS, and AMS is the cost of instruments.

In addition, the application of radionuclides in tracing has become a hot and difficult issue. Using the atomic ratio of multiple elements, such as ^90^Sr/^137^Cs, ^134^Cs/^137^Cs, ^240^Pu/^239^Pu, etc., to find a more accurate source of pollution has been applied to the study of the migration of sand and dust around Japan and Asia, including the condition of melted fuel in the damaged core after the accident. The application of tracing is of great significance to further explore the impact of nuclear accidents and nuclear waste emissions on human health and the environment. The application of ^137^Cs and other radionuclide tracing techniques in aerosols has been seldom studied, and it has yet to be reported by relevant researchers.

## Figures and Tables

**Figure 1 ijerph-19-10183-f001:**
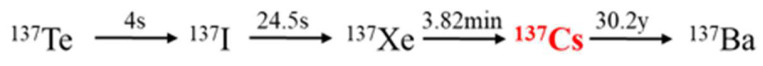
The decay chain of ^137^Cs.

**Table 1 ijerph-19-10183-t001:** Interferences affecting the determination of ^137^Cs^+^ by mass spectrometry, reprinted with permission from [49].

Analyte	Interference	Abundance (%)
^137^Cs^+^	^137^Ba^+^	^137^Ba 11.3
	^136^Ba^1^H^+^	^136^Ba 7.81
	^136^Xe^1^H^+^	^136^Xe 8.87
	^121^Sb^16^O^+^	^121^Sb 57.2
	^97^Mo^40^AR^+^	^97^Mo 9.6

**Table 2 ijerph-19-10183-t002:** Performance parameters of ^137^Cs.

Sample	Pretreatment	Separation	Measurement Method	Recovery (%)	Detection Limit	Ref.
Soil	Acid leaching	AMP + AG 50W-X8	CRC-ICP-MS/MS	-	0.06 ppt	[50]
Soil	Fusion with NaOH and Na_2_O_2_	AMP-PAN	TIMS	-	-	[26]
	Fusion with Li_2_B_4_O_7_ and LiBO_2_			-	-	
Soil, plant	Ashed	-	γ-ray spectrometry	-	-	[7]
Soil	Ashed + acid leaching	AMP + AG MP-1 M + AG 50W-X8	ICP-MS/MS	>95	0.006 pg·mL^−1^	[55]
Soil, sludge, sediment	Ashed + fused with Li metaborate and LiI	AMP	ICP-MS	-	0.09 ng·L^−1^	[28]
Soil	Ashed + acid leaching	AMP-PAN + AG 50W-X8	TIMS	-	0.13 mBq·L^−1^	[67]
Soil, sediment	Ashed + acid leaching	AMP-PAN + AG 50W-X8	γ-ray spectrometry	60%	-	[68]
	Ashed + fusion with LiBO_2_			93%	-	
Sediment	-	-	γ-ray spectrometry	-	0.8 mBq·g^−1^	[56]
Sea Sediment	Aqua regia acid leach and filtration/centrifuge	AMP + AG 50W-X8	ICP-SFMS	78 ± 3	0.05 ng·kg^−1^	[27]
	Acid leaching in ultrasonic bath + evap			80 ± 6		
	Lithium metaborate fusion			100 ± 6		
	Single attack NaOH sinter in silica crucible			90 ± 9		
	Double attack NaOH sinter in silica crucible			100 ± 9		
	Acid leach + NaOH sinter			100 ± 10		
Seawater	Acidified	AMP-PAN	γ-ray spectrometry	85–94	0.15 Bq·m^−3^	[69]
		KniFC-PAN		93		
Coastal water	Acidified	AMP + AG 50W-X8	ICP-MS	100.1 ± 3.3	1.0 ng·L^−1^	[70]
Seawater	-	AMP + Doulite C-3	β-counting	90	-	[14]
Seawater	Acidified	KniFC−PAN	γ-ray spectrometry	99	-	[71]
Seawater	Acidified	AMP	γ-ray spectrometry	-	-	[1]
Seawater	Acidified	KniFC−PAN	γ-ray spectrometry	87–99	-	[72]
Seawater	-	AMP	γ-ray spectrometry	-	-	[73]
Lake water	-	Adsorbed by a thin film of mixed ferrocyanide of potassium iron	γ-ray spectrometry	50–90	-	[74]
Seawater	Acidified	AMP	γ-ray spectrometry	-	0.20 Bq·m^−3^	[75]
Seawater	Acidified	AMP-PAN + AG 50W-X8	TIMS	-	0.13 mBq·L^−1^	[67]
Aerosol	-	-	γ-ray spectrometry	98–99	-	[76]
Aerosol	-	-	γ-ray spectrometry	-	-	[77]
Oyster	Irradiation	-	γ-ray spectrometry	-	-	[34]
Skull	Ashed	-	γ-ray spectrometry	-	-	[33]
Food	Ashed	-	γ-ray spectrometry	-	-	[44]
Leaf, litter	Ashed + acid leaching with H_2_O_2_	AMP + AG MP-1 M + AG 50W-X8	ICP-MS/MS	>95	0.006 pg·mL^−1^	[55]

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
