# Peer review of "Determination, Separation and Application of 137Cs: A Review"

_ijerph, 2022, doi:10.3390/ijerph191610183_

Round 1

Reviewer 1 Report

This paper present a good review on the analytical method for 137Cs in the environment, the paper was well organized and presented, the topic is interesting for the readers in the field of environmental radioactivity and radiation protection. It is recommended to be published as it is or a minor revision.

1. Fig. 2 is not informative and necessary, suggest to delete it.

2. "3. Determination" should be written as "Measurement", determination include the sample preparation and separation of target radionuclides, but this part only focus on the measurement methods. 

3. Table 2, the reference should be in the last column.

4. "5. research of the results calculation" should be written ase "evaluation of measurement data".

5. "Study on 137Cs as tracer in ocean transportation and convex" should be written as " Application of 137Cs as oceanographic tracer"

6. "Study on the deposition of137Cs as tracer in soil" should be written as "Application of global fallout 137Cs as tracer for soil erosion investigation".

7.  In the references list, the reference after No. 50 should be numbered. 

Reviewer 2 Report

137Cs is among the most important hazardous radionuclides involved in radiological pollution to human and the environment due to their long half-lives and high fission yields. The authors reviewed the latest research progress of radioactive cesium from five aspects, including sample pretreatment, determination, separation, calculation, application of radioactive cesium. However, there are some problems that need to be concerned in this manuscript. Therefore, I suggest the minor revisions before publication. Some specific key points are as follows:

1. The manuscript should be thoroughly checked for English corrections as there are some colloquial terms being used. Some typing should be corrected, including but not limited to the following examples.

Line 26, “545-765 pBq 137Cs” should be corrected to “545-765 PBq 137Cs”.

Line 29, “85 pBq 137Cs into the surrounding environment” need to “85 PBq 137Cs into the surrounding environment”.

2. The authors need to take note in the revision stage and cite relevant references including high impact journal to make the manuscript in broad range readers.

3.Please include full page ranges for references 20, 22, 29, 51, 53, 56, 101. If these references refer to article numbers, rather than page numbers, the inclusion of the article number without extra text, e.g., “article no”, is fine.

4. Abbreviations that appear for the first time in the article need to be given their full names, including ICP-MS, ICP-QMS, SF-ICP-MS, DRC-ICP-MS, ETV-ICP-MS.

5. The quality of figures should be improved. The size of the figure legend should be enlarged. The legend for Figure 2 (Line 59) should be placed in the correct position.
